# Multi-Objective Lower Irrigation Limit Simulation and Optimization Model for *Lycium Barbarum* Based on NSGA-III and ANN

**Jinpeng Zhao, Yingduo Yu \*, Jinyang Lei and Jun Liu**

Department of Irrigation and Drainage, China Institute of Water Resources and Hydropower Research,
National Center for Efficient Irrigation Engineering and Technology Research, Beijing 100089, China
\* Correspondence: yuyingduo_email@163.com; Tel.: +86-19530104881

**Abstract:** *Lycium barbarum* has rich medicinal value and is an important medicinal and economic tree species in China, with an annual output value of 21 billion RMB. The yield and the quality of *Lycium barbarum* dry fruit are the crucial issues that affect the cultivation of *Lycium barbarum* and the income of farmers in the Ningxia water shortage area. According to the local acquisition standard of *Lycium barbarum*, the amount of dry fruit per 50 g (ADF-50) is the key factor in evaluating the quality and determining the purchase price. In order to optimize the irrigation lower limit of automatic drip irrigation system with multiple objectives, the yield and ADF-50 are selected to be optimal objectives. The lower irrigation limits of the automatic drip irrigation system in the full flowering stage, the summer fruiting stage, and the early autumn fruiting stage are optimized by the third generation of non-dominated sorting genetic algorithm (NSGA-III) in this paper. The mathematical relationships between irrigation lower limit and irrigation quantity, irrigation amount, yield, and ADF-50 were established by the water balance model, water production function (WPF), and artificial neural network model (ANN), respectively. The accuracy of the water balance model and ANN were verified by experiments. The experiments and optimization results show that: (1) irrigation quantity and ADF-50 calculated by the water balance model and ANN are accurate, and their Nash–Sutcliffe coefficient are 0.83 and 0.66; (2) In a certain range of irrigation quantity, ADF-50 and *Lycium barbarum* yield show competitive relation. By solving the NSGA-III optimization model, the lower irrigation limits schemes, which tend to different objectives, and a compromise scheme can be obtained; (3) Compared with the original lower limit of irrigation water, the compromise scheme's yield and quality of *Lycium barbarum* are improved 10.7% and 8.8% respectively. The results show that the automatic drip irrigation system's lower irrigation limit scheme optimized by the model can improve not only the yield but also the quality of *Lycium barbarum*. This provides a new idea for establishing the irrigation lower limit of the automatic drip irrigation system in the *Lycium barbarum* planting area.

**Keywords:** *Lycium barbarum*; NSGA-III; ANN; multi-objective optimization; lower irrigation limits

## 1. Introduction

*Lycium barbarum* is an important commercial crop and medicinal food herb in the Ningxia autonomous region of China and has high medicinal value. Adding *Lycium barbarum* to a regular diet can effectively nourish one's liver and kidneys [1–3]. Ningxia autonomous region is a continental arid climate, and its annual precipitation does not exceed 400 mm. The shortage of irrigation water resources has become the main reason that limits the development of the *Lycium barbarum* industry. Under limited water resources, the optimization of the irrigation system for *Lycium barbarum* can further reduce irrigation losses while maintaining the yield and quality at the same time, which is of great significance to arid areas in western China. A large number of scholars have studied the relationship between the irrigation system and the yield and quality of *Lycium barbarum* [4–7], but most

of the existing research chose the better irrigation scheduling by comparing the different irrigation treatments in plot experiments, and these were mostly conducted by manual control irrigation [8–10]. Using the schemes comparison to optimize the irrigation system has the following two disadvantages: (1) The experiment period is too long; (2) When there are too many factors to be optimized in the irrigation scheduling, it is bound to cause a large gap between treatments and affect the optimization accuracy. With the improvement of automation technology, there are an increasing number of *Lycium barbarum* plantations changing to use automated irrigation systems. However, there is little research on the optimization of automatic irrigation systems.

In recent years, more scholars have adopted the optimization–simulation coupling model to optimize the research object [11–15]. The optimization–simulation coupling model can improve not only the optimization efficiency but also the accuracy in a short time through a large amount of experiment simulation [16,17]. With the increasing number of objectives in optimization problems, more researchers use the non-dominated sorting genetic algorithm (NSGA) as an optimization model in a coupling model, which can deal with multiple competing objectives well. For example, Liu et al. used NSGA to optimize irrigation scheduling under different precipitation and evaporation conditions with the aim of maximizing water production efficiency and yield [18]; In order to solve the contradiction between energy consumption and crop yield of the pressurized irrigation network, M.T.Carrillo Cobo adopted NSGA to optimize the irrigation pattern of pressurized irrigation network [19]. But so far, there is no study on the optimization of the lower limit of automatic drip irrigation of *Lycium barbarum* with NSGA. The third generation of the NSGA algorithm is introduced into the lower irrigation limit optimization problem of automatic drip irrigation of *Lycium barbarum* to solve this multi-objective optimization problem more efficiently. However, the use of the simulation–optimization coupling model to optimize the irrigation system of *Lycium barbarum* needs to establish the mathematical relationship between irrigation quantity, yield, and quality. But the optimization object in this study is the lower irrigation limit of the automatic drip irrigation system, so we need to establish the mathematical relationship between lower irrigation limit and quantity. A water balance model is a good choice. The water balance model has characteristics of convenient calculation and needs fewer correlation parameters. Liu et al. established the water balance model to simulate water transport in an irrigation area and got accurate simulation results [20]. So, a *Lycium barbarum* active root layer water balance model was established to calculate irrigation amount according to the irrigation upper and lower limits and water content.

Besides, most current studies established the water production function of crops, including *Lycium barbarum* [21–23], but there was no research on the mathematical relationship between *Lycium barbarum* quality and irrigation scheduling. In this study, the quality of *Lycium barbarum* was measured by ADF-50, but the influence mechanism of irrigation amount on ADF-50 remains unclear. With the development of artificial intelligence in recent years, Artificial Neural Networks (ANN) are widely used in the construction of various prediction models or simulation models. For example, Kasiviswanathan et al. developed an ANN to forecast the weekly reservoir inflows along with its uncertainty, and the prediction precision is good [24]; Saber et al. created an accurate and reliable ANN model for irrigation parameters to predicate irrigation water quality [25]. When using the ANN model to construct the underlying mathematical relationship between parameters, it is not necessary to define the physical relationship between parameters, and compared with the widely used method of solving the functional relationship through the least square method, the neural network model does not need to clarify the functional form of the mathematical relationship in advance [26,27]. Therefore, in this study, the neural network model is used to establish the mathematical relationship between ADF-50 and irrigation quantity.

After all the simulation models have been established, the ADF-50 neural network model, the water content simulation model of active root layer, and the soil water production function were embedded into the genetic algorithm as the objective functions.

The smaller the ADF-50, the quality of Lycium berry is better. Therefore, the maximum yield and the minimum ADF-50 were taken as the objectives, and the lower irrigation limit was taken as the decision variable. A simulation–optimization coupling model based on a neural network and multi-objective genetic algorithm model NSGA-III was used to optimize the irrigation limit of the automatic drip irrigation system in the *Lycium barbarum* planting area. Previous studies have shown that the maximum yield of *Lycium barbarum* and the minimum ADF-50 targets cannot be reached at the same time [28], and there are few studies on automated drip irrigation schemes that can balance the yield and quality of *Lycium barbarum*. Therefore, this study aims to establish a reasonable irrigation lower limit scheme which tends to different objectives for the automated drip irrigation system of *Lycium barbarum* by the NSGA-III algorithm. In addition, we obtain a compromise scheme by assigning equal weight to different objective function values.

## 2. Materials and Methods

### 2.1. Overview of the Study Area

The experiment was conducted from March 2018 to October 2019 in Ningxia Zhongwei Jiusheng Agricultural Park (105°06′ E 37°27′ N) at the intersection of Ningxia, Inner Mongolia, and Gansu provinces in the middle and upper reaches of the Yellow River. Its altitude is 1231 m, and this region is a typical temperate continental monsoon climate-perennial droughty, less rainy, adequate sunshine, and widely varies temperature from day to night. The effective annual precipitation was about 147 mm, mainly in July and August, accounting for 57.23% of the year. The annual average temperature was 9.2 °C. The annual average sunshine duration was 2728.0 h, and the annual evaporation was 1921 mm. The frost-free period was about 150 days. The wind speed was about 2–6 m/s, and the frozen depth of soil in the park was about 1 m. The soil in the park is sandy loam with porosity of 47% and field water capacity of 19.8%. In this study, 10 sites were selected (Figure 1).

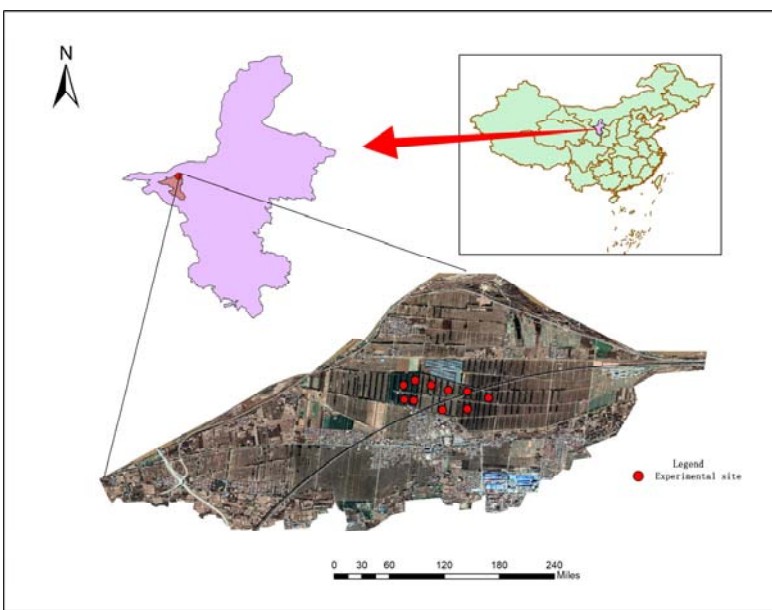

**Figure 1.** Study area location and experimental sites within PR China.

### 2.2. Lycium Barbarum's Active Root Layer Water Balance Model

Although the water balance model cannot reflect the groundwater migration process, it is widely used because of its high calculation efficiency. So, the principle of water balance was used to simulate the change process of water content in the active layer of *Lycium barbarum's* root in this study. The field water capacity was taken as the initial water content of the active root layer system. The water balance method was used to calculate the water content of the active root layer system day by day, and the calculation formula was shown in

Equation (1). When the water content was lower than the lower limit of irrigation, irrigation times and irrigation quota of the whole growth period of *Lycium barbarum* were recorded.

$$\theta_i h = \theta_{i-1} h + P_i + I_i - K_s ET_0 - Q_i \tag{1}$$

where, $\theta_i$ and $\theta_{i-1}$ are the water content of active root layer per unit area on day *i* and day *i-1*, respectively; *h* is the thickness of active root layer (mm). $P_i$ is the rainfall on day *i* (mm). $I_i$ is the irrigation volume (mm) on day i; $K_c$ is soil moisture coefficient; $ET_0$ is the reference crop exfoliation (mm), the Penman–Monteith model recommended by the Food and Agriculture Organization of the United Nations (FAO) used to calculate the reference crop exfoliation in this study; $Q_i$ is the sum of groundwater leakage and recharge in the active root layer (mm).

Because drip irrigation was used in this study and the buried depth of the groundwater level was 20 m, groundwater leakage and recharge are ignored in this model. In addition, in order to simplify the simulation process, when the water content in the active root layer exceeds the water content in the field due to rainfall, the model considers that the excess water will be discharged on the same day, and the water content in the active root layer at this time is the water content in the field. In this study, the Nash–Sutcliffe coefficient was used to assess the water balance model (Equation (2)).

$$E = 1 - \frac{\sum_{t=1}^{T} \left( A_0^t - A_m^t \right)^2}{\sum_{t=1}^{T} \left( A_0^t - \overline{A_0} \right)^2} \tag{2}$$

where, $A_0^t$ is experiment results; $A_m^t$ is simulation results; $\overline{A_0}$ is the average of experiment results.

The closer the Nash–Sutcliffe coefficient is to 1, the better the simulation effect is.

### 2.3. Water Production Function of Lycium barbarum

The water production function of *Lycium barbarum* is an expression to describe the mathematical relationship between the yield of *Lycium barbarum* and the amount of irrigation water. The expression of the water production function is shown in Equation (3) [29], and the correlation coefficient r = 0.9547.

$$Y = -0.00025x^2 + 0.39276x - 17.2013 \tag{3}$$

where Y is yield; x is the irrigation water amount in the whole growth period of *Lycium barbarum*.

When the irrigation quantity in the whole growth period of *Lycium barbarum* was calculated by the water balance model, the irrigation quantity was brought into expression (2) to calculate the yield.

### 2.4. ADF-50 Artificial Neural Network Model

The artificial neural network model is a mathematical model that simulates the structure of the neural network in the brain. It consists of an input layer, a hidden layer, and an output layer (Figure 2). After being trained by historical data, a neural network with a specific mathematical relationship between the input value and output value is established. In this study, back propagation artificial neural network (BP-ANN) was used to construct the mathematical relationship between irrigation quantity and ADF-50; BP-ANN is a multilayer feedforward neural network trained according to an error-backward propagation algorithm. The connection weight between each hidden layer was adjusted according to the error between the measured value and the output value until the error was less than the allowed value. The training steps of the BP-ANN were as follows:

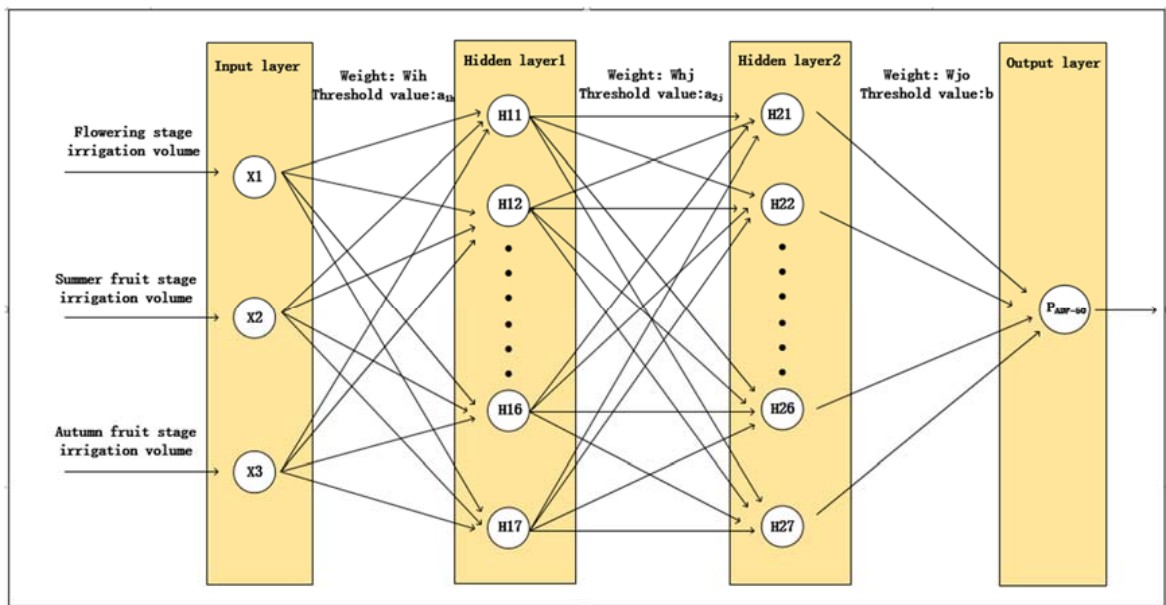

**Figure 2.** Structure diagram of ADF-50 neural network model.

Step 1: training sample expansion. A large number of training samples are needed to ensure the accuracy of the BP-ANN model. But the amount of ADF-50 data was insufficient, so in addition to the data obtained from the experiment, the farmers of *Lycium barbarum* with automatic drip irrigation systems and planted with the same variety and tree age were investigated. A total of 50 groups of experimental data were obtained. 10 of the data groups were used to verify the BP-ANN's accuracy. In order to further make up for the effect of lacking training data, the remaining 40 groups data were expanded [30], and 400 data samples were generated.

Step 2: initialize the neural network. The input value was the irrigation amount of Lycium berry in the full flowering stage (T1), summer fruit stage (T2), and early autumn fruit stage (T3), and the output value was ADF-50. Two hidden layers were set, and the number of nodes in the input layer, hidden layers, and output layer were N = 3, L = 7, and M = 1, respectively. The initial weights $W_{ih}$, $W_{hh}$, and $W_{ho}$ of each layer were set as 0.5, the learning rate was 0.1, the initial thresholds a and b for the hidden and output layers was 0.3, and the sigmoid function was selected as the activation function. The expression is shown in Equation (4).

$$f(x) = \frac{1}{1 + e^{-x}} \tag{4}$$

Step 3: calculate the hidden layer's output. The output was calculated by input variables *X*, weights $W_{ih}$, $W_{hj}$, and the threshold $a_i$. The calculation formulas are shown in Equations (5) and (6).

$$H_{1h} = f(\sum_{i=1}^{N} w_{ih} X_i - a_{1i})\, h = 1, 2 \cdots L \tag{5}$$

$$H_{2j} = f(\sum_{h=1}^{L} w_{hj} H_{1h} - a_{2h})\, j = 1, 2 \cdots L \tag{6}$$

Step 4: calculate the output layer's output value of the output layer. Based on the output value of the No. 2 hidden layer $h_{2j}$, the connection weights $W_{jo}$ and the threshold, the output value $P_{ADF-50}$ was calculated. The calculation formula is shown in Equation (7).

$$P_{ADF-50} = f\left(\sum_{j=1}^{L} w_{jo}H_{2j} - b_j\right) \tag{7}$$

Step 5: calculate the error between the ADF-50 measured value and the neural network calculated value, and the calculation formula is shown in Equation (8).

$$e = A_{ADF-50} - P_{ADF-50} \tag{8}$$

Step 6: update weights and thresholds. According to error $e$, the weights and thresholds were updated, and the formulas are shown in Equations (9)–(14).

$$W_{jo} = W_{jo} + \alpha H_{2j}e \tag{9}$$

$$W_{hj} = W_{hj} + \alpha H_{2j}(1 - H_{2j})H_{1h}W_{hj}e \tag{10}$$

$$W_{ih} = W_{ih} + \alpha H_{1h}(1 - H_{1h})X_i W_{ih}e \tag{11}$$

$$b = b + e \tag{12}$$

$$a_{2j} = a_{2j} + \alpha H_{2j}(1 - H_{2j})W_{jo}e \tag{13}$$

$$a_{1h} = a_{1h} + \alpha H_{1h}(1 - H_{1h})\sum_{j=1}^{L} W_{hj}e \tag{14}$$

Step 7: if the error was less than the allowed value (10%) or the pre-defined training time was reached, the training would stop. If the conditions were not reached, return to Step 3. After training, the network was able to calculate ADF-50.

The Nash–Sutcliffe coefficient was also used to assess the ADF-50 artificial neural network model.

*2.5. Multi-Objective Genetic Algorithm Optimization Model NSGA-III*

In this study, the multi-objective genetic algorithm optimization model NSGA-III was used to optimize the lower irrigation limits of the automated drip irrigation system in the *Lycium barbarum* field. The model optimized the decision variables by simulating the biological evolution process, and the optimization was an iterative process in essence. The combination of decision variables to be optimized was regarded as biological individuals. When the iteration started, the population size was set, and the decision variables were initialized to form the first-generation population. By simulating the evolution process of the biology, the evaluation of each individual was calculated by the objective function; individuals that had better evaluations were selected for crossover and mutation operation so as to generate a new generation of population. Each iteration generated a new generation until the evaluation value reached the preset convergence standard.

The parameters of the NSGA-III optimization model were set as follows: population size N = 200, iterations G = 100, binary crossover and mutation were used, and the crossover and mutation probability were 0.6 and 0.01, respectively. If there was competition between the objective functions, it often could not get a single global optimal solution when the NSGA-III was used to solve the multi-objective problem, but a non-dominated set included solutions that tended to different targets. These solutions performed better on their preferred goals while ensuring that the other goals were not too bad.

### 2.5.1. Decision Variable

Three decision variables were the irrigation lower limits of automatic drip irrigation in three growth stages: full flowering, summer fruit, and early autumn fruit stage. The setting of decision variable parameters is shown in Equation (15).

$$L_i, \ i = 1, 2, 3 \tag{15}$$

where: $L_i$ is the irrigation lower limits of automatic drip irrigation in three growth stages.

### 2.5.2. Objective Function

This study included two objective functions: maximum yield and minimum ADF-50, as shown in Equation (16).

$$\begin{cases} \max Z_1 = -0.00025 \left( \sum_t^T Q_t \right)^2 + 0.39276 \left( \sum_t^T Q_t \right) - 17.2013 \\ \min Z_2 = f_{ANN} \left( \sum_t^T Q_t \right) \end{cases} \quad t = 1, 2 \dots 5 \tag{16}$$

where: $f_{ANN}$ is the mathematical relationship model between irrigation volume and ADF-50 constructed by a neural network; $Q_t$ is the irrigation amount of Lycium Barbarum at each growth stage, which is respectively the irrigation amount at the spring shoot stage, full flowering stage, summer fruit stage, early autumn fruit stage, and late autumn fruit stage.

### 2.5.3. Constraint

Considering the actual irrigation demand, the lower limit of automatic drip irrigation should not be lower than the wilting coefficient and higher than the upper limit of irrigation water. When rainfall occurs, the water content of the active root layer may exceed the field water capacity. However, in order to simplify the water balance simulation process, it was assumed that the water exceeding the field water capacity was discharged from the active root layer on the same day, and the water content of the active root layer $\theta$ did not exceed the field water capacity. The constraint conditions are shown in Equation (17).

$$\begin{cases} \theta_w \leq \ L_i \ \leq 0.95 \ \theta_f \\ \theta \leq \theta_f \end{cases} \tag{17}$$

where: $\theta_f$ is field water capacity. In this study, the upper limit of irrigation was set at 95% $\theta_f$; $\theta_w$ is wilting coefficient; $\theta$ is the water content of the active root layer, $L_i$ is the lower limit of drip irrigation water in the optimized three growth stages.

### 2.6. Construction of the Coupling Model

The NSGA-III optimization model calculated ADF-50 and yield by transferring the ADF-50 ANN model, water production function, and water balance model of the *Lycium barbarum* active root layer. Before the optimization, the ADF-50 ANN model needed to be trained. The neural network model updated the threshold and weight by reading the training samples until the error between the calculated result and actual result was less than the allowed value. The potential mathematical relationship between ADF-50 and irrigation volume could be established through the trained thresholds and weights. After the neural network model was trained, NSGA-III could be used to optimize the lower limit of drip irrigation. The optimization process was a cyclic process. When the cycle started, the population size and iterations were set, and the decision variables were initialized to form the first-generation population. Each feasible solution (lower limit of drip irrigation water) in the population was brought into the water balance model of *Lycium barbarum* active root layer to calculate the irrigation amount, and then the irrigation amount was input into the neural network model and water production function to calculate the ADF-50 and yield. The feasible solutions were non-dominated and sorted according to the yield and ADF-50.

The individuals at the top of the sort were selected to cross over and mutate to generate new individuals, which were merged with the previous generation's population to obtain a new population. The above steps were repeated until the maximum number of iterations or convergence conditions were reached. The construction flow chart of the coupling model is shown in Figure 3.

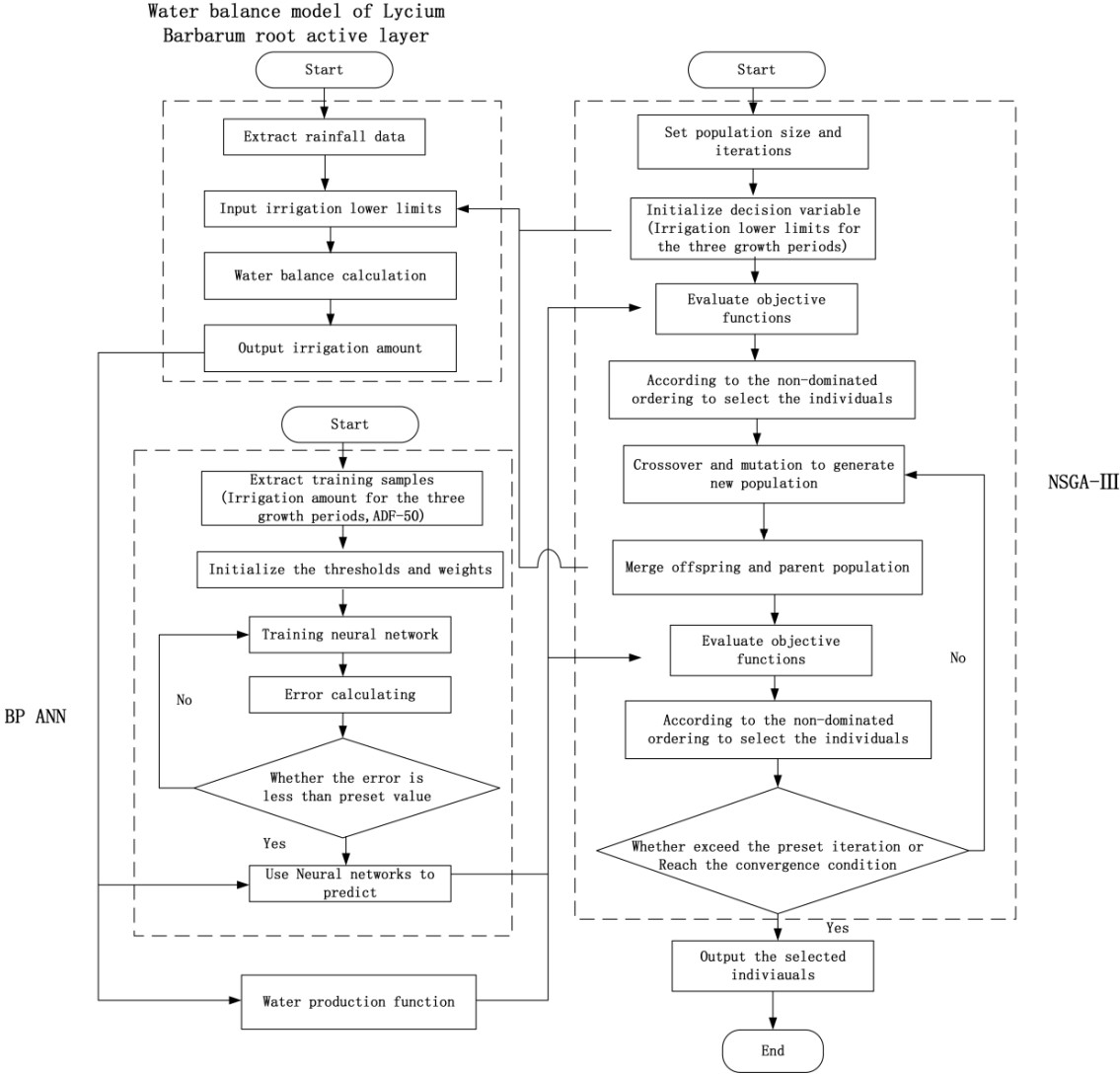

**Figure 3.** Construction flow chart of the coupling model.

## 3. Results

### 3.1. Simulation Models Validation

The results of the water balance model and ADF-50 neural network model in the Lycium barbarum active root layer were verified. The experimental data (lower limit of drip irrigation water at three stages) that were not used as the neural network training samples were input into the water balance model to calculate the irrigation quantity. The measured and simulated values are shown in Figure 4. Then input the irrigation quantity to the ADF-50 neural network model to calculate the ADF-50. The calculation results and the measured values are shown in Figure 5. The Nash–Sutcliffe coefficient of the water balance model and ADF-50 neural network model were 0.83 and 0.66. The nearer to 1 the Nash–Sutcliffe coefficient approaches, the higher the simulation accuracy will be; if it is much less than 0, the simulation results of the model are unreliable. Although ADF-50 neural network model had a slightly lower Nash–Sutcliffe coefficient, the simulation accuracy of

these two models was acceptable. The verification results of simulation models showed that the water balance model could simulate the water content of the active root layer and calculate irrigation quantity well. At the same time, ADF-50 neural network model could calculate ADF-50 accurately.

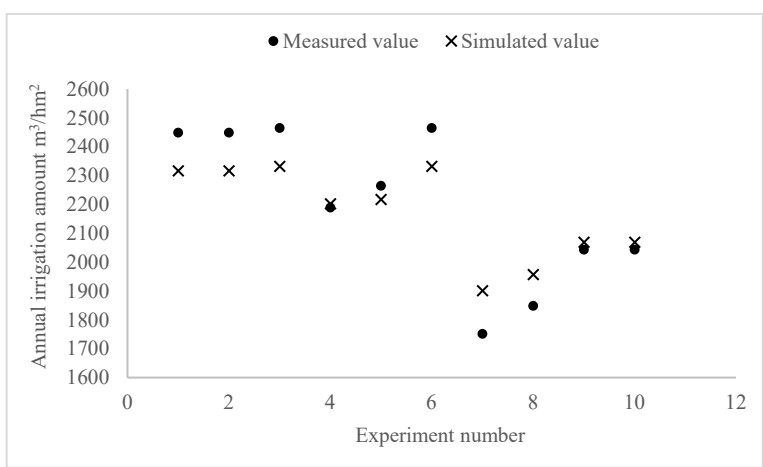

**Figure 4.** Verification of water balance model in *Lycium barbarum* active root layer.

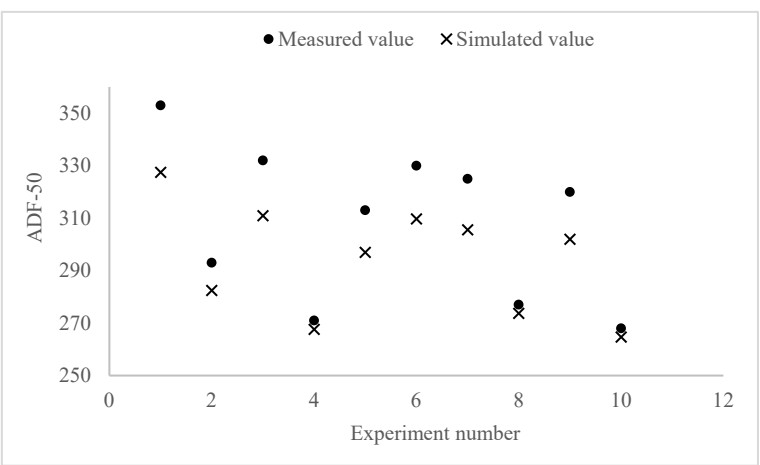

**Figure 5.** ADF-50 neural network model verification.

### 3.2. Optimization Results of Irrigation Drip Lower Irrigation Limit for Lycium barbarum

The optimization purpose of this study was to obtain the lower drip irrigation limit schemes that tended to different objectives or compromise between the two objectives. The simulation and optimization coupling model was used under the conditions of precipitation and evaporation in 2018 and 2019, respectively. Each objective function value of the lower irrigation limit scheme of drip irrigation was counted (Figures 6–8). In these figures, the larger the dot was, the scheme performed better in this objective. In order to select the scheme that had compromised performance, the sum of two objective function values was used as the scheme's score after being given the same weight and normalization. When selecting the scheme based on the score, the lower irrigation limits at three growth stages were about 65%, 50%, and 65% (Percentage of field capacity), respectively. The maximum yield of Lycium barbarum was achieved when the lower limits were about 70%, 50%, and 70% in full flowering, summer fruit, and early autumn fruit stage, respectively. The lower irrigation limits in the full flowering and early autumn fruit stage had little effect on ADF-50, while the most favorable lower irrigation limit at the summer fruit stage was about 50%.

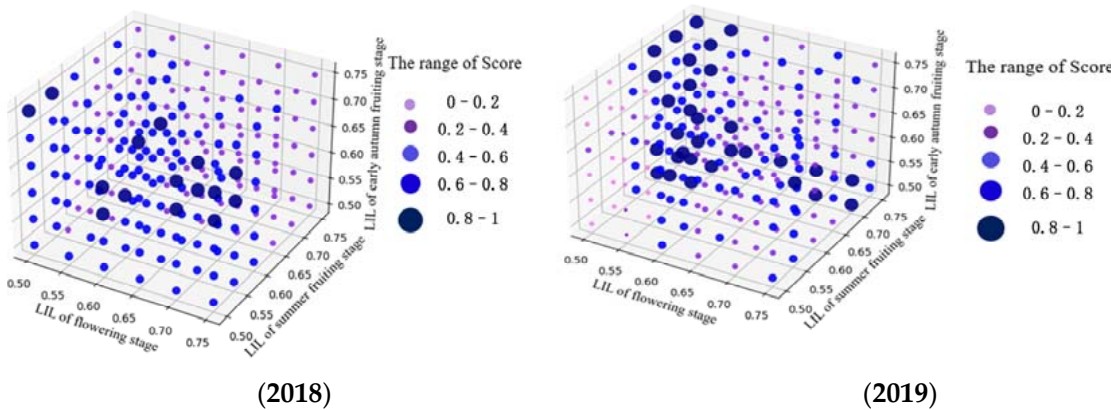

**(2018)** **(2019)**

**Figure 6.** Scatterplot of relationship between lower irrigation limit and score at different growth stages (LIL means lower irrigation limit).

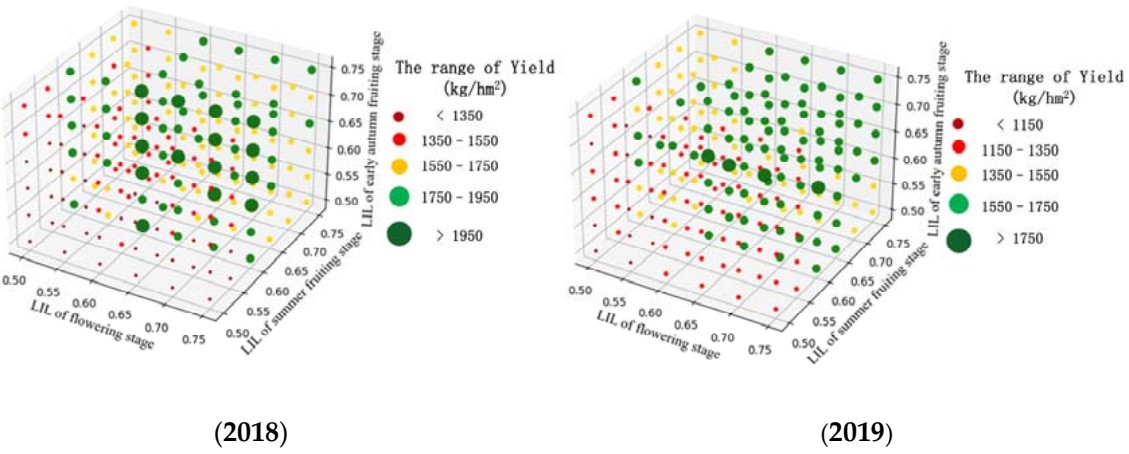

**(2018)** **(2019)**

**Figure 7.** Scatterplot of relationship between lower irrigation limit and yield at different growth stages.

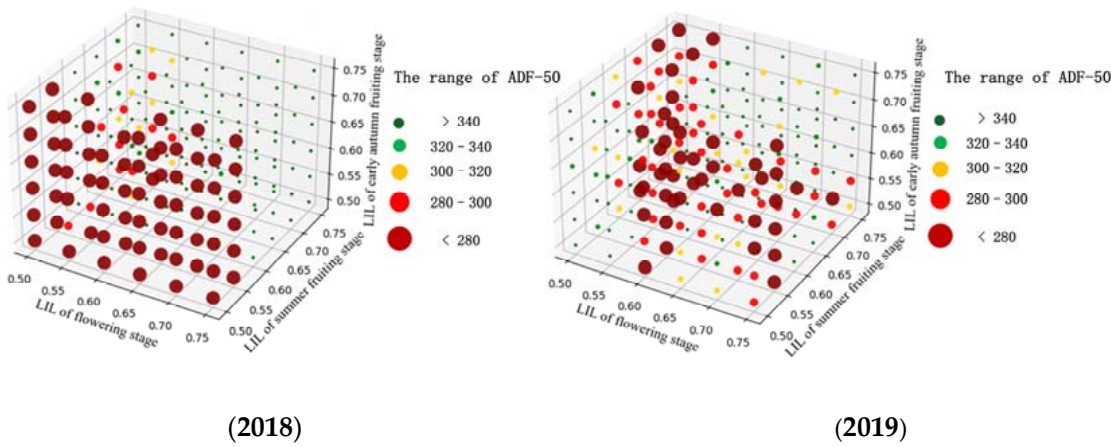

**(2018)** **(2019)**

**Figure 8.** Scatterplot of relationship between lower irrigation limit and ADF-50 at different growth stages.

Three typical irrigation schedulings that tended to different goals were selected from the non-dominated set (Table 1), in which the S1 scheme was the compromise scheme that balances the two goals, the S2 scheme was the scheme that tended to yield, and the S3 scheme was the scheme that tended to ADF-50.

**Table 1.** Functions' values for the 3 treatments.

| Scheme | Lower Irrigation Limits (Percentage of Field Capacity) | | | Yield (kg/hm$^2$) | | ADF-50 | |
|---|---|---|---|---|---|---|---|
| | T1 | T2 | T3 | 2018 | 2019 | 2018 | 2019 |
| S1 | 65% | 50% | 65% | 1655.21 | 1473.34 | 312 | 306 |
| S2 | 70% | 50% | 70% | 1983.69 | 1757.82 | 357 | 352 |
| S3 | 60% | 50% | 65% | 1322.95 | 1271.46 | 247 | 249 |
| Original | 50% | 60% | 65% | 1473.76 | 1348.87 | 342 | 336 |

Notes: T1 means the full flowering stage, T2 means the summer fruit stage, and T3 means the early autumn fruit stage.

The irrigation times and irrigation quantity of the lower irrigation limit schemes that tended to different objectives were statistically analyzed (Figure 9). The irrigation times from more to less in proper order were S2, original scheme, S1, and S3. The trend of the four schemes' total irrigation water volume was consistent with irrigation times. In 2018, the irrigation time and quantity of S1 were 21 and 1945.06 m$^3$/hm$^2$; S2 were 25 and 2628.06 m$^3$/hm$^2$; S3 were 13 and 1648.10 m$^3$/hm$^2$; original scheme were 22 and 2405.34 m$^3$/hm$^2$. In 2019, the irrigation time and quantity of S1 were 23 and 2152.93 m$^3$/hm$^2$; S2 were 28 and 2910.16 m$^3$/hm$^2$; S3 were 14 and 1781.73 m$^3$/hm$^2$; original scheme were 25 and 2687.45 m$^3$/hm$^2$. Based on the analysis of the lower limit of irrigation water in Table 1, it could be seen that the irrigation times and the total quantity of irrigation water were positively correlated with the lower limit of irrigation water. In addition, within the constraint range, the yield and ADF-50 showed an increasing trend with the increase of the total irrigation quantity, indicating that increasing the lower limit of irrigation water and increasing the irrigation water could improve the yield but reduce the quality of *Lycium barbarum*. It is worth noting that the larger the ADF-50, the worse the quality. Compared with the original lower limit scheme of drip irrigation, the S1 scheme with compromise consideration of the two objectives could increase the yield by 10.7% while reducing the ADF-50 by 8.8% and improving the quality of *Lycium barbarum* while increasing the yield. The S2 increased yield by 32.5% at the cost of increasing ADF-50 by 4.6%. In contrast, the S3 decreased ADF-50 by 26.8% at the cost of decreasing yield by 8.0%.

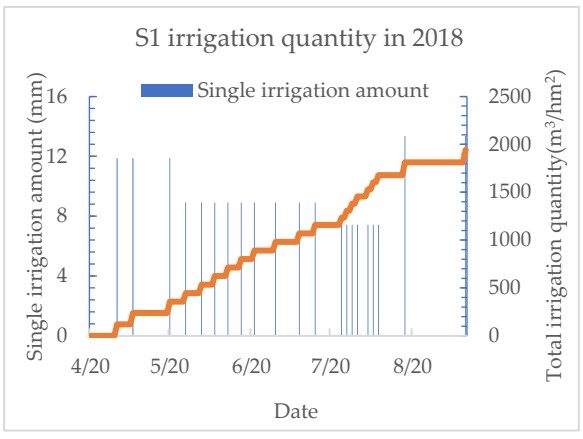
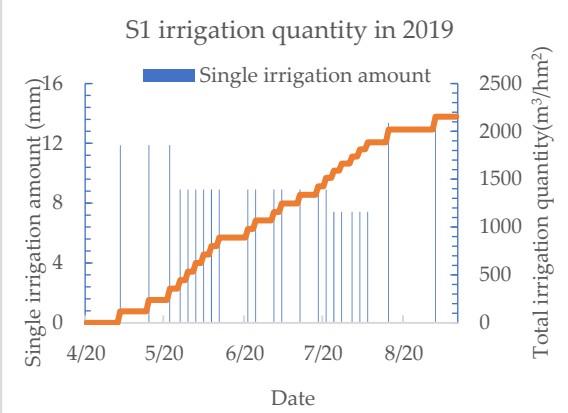

**Figure 9.** *Cont*.

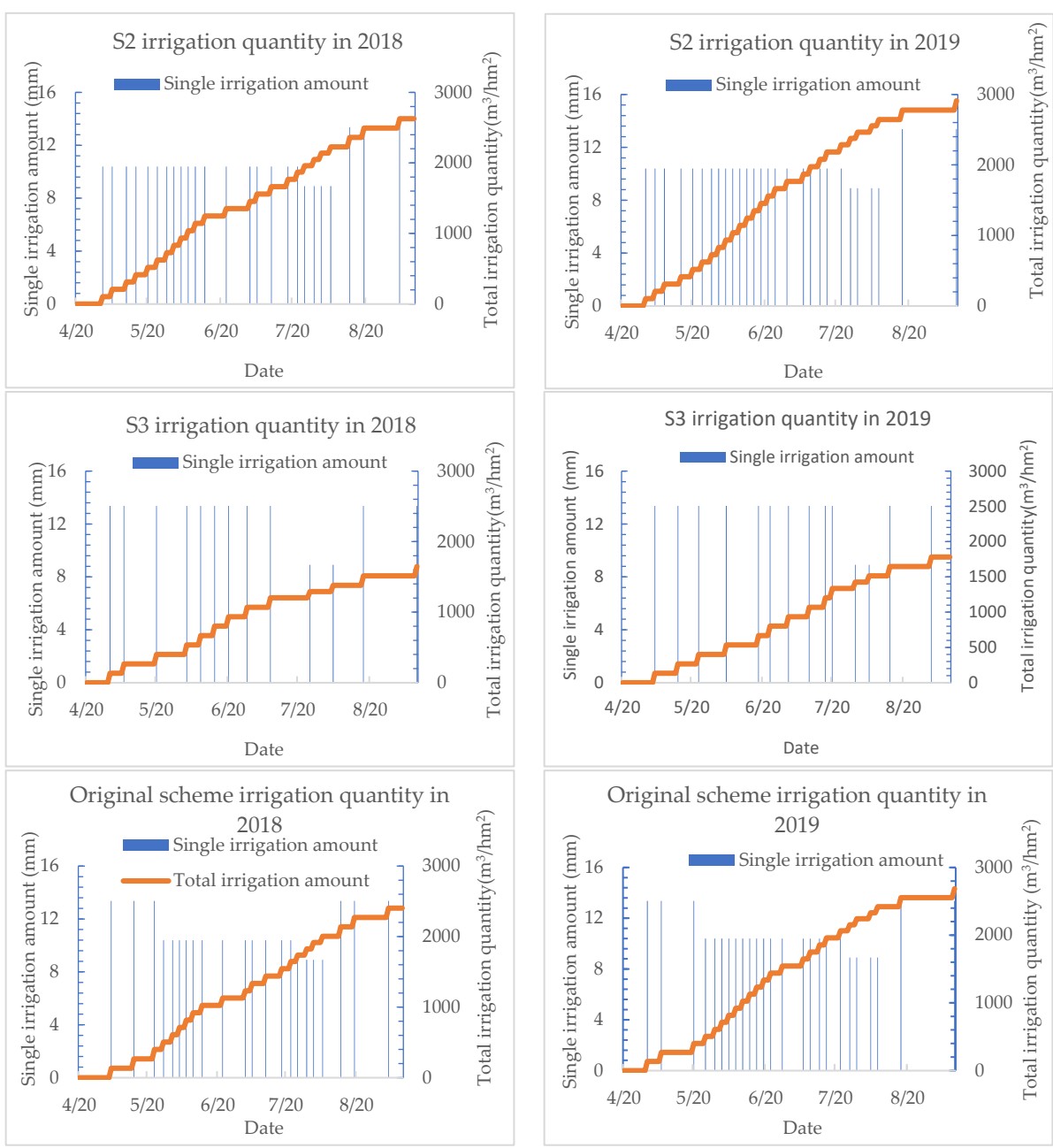

**Figure 9.** Irrigation distribution of different LIL solutions.

In conclusion, these results demonstrate that the simulation models can be used as objective functions in the NSGA-III optimization model because of their high accuracy. Further, within the lower limits of irrigation constraints, the two objectives have a competitive relationship. Optimizing one objective is at the expense of the other. Farmers can choose different schemes according to their preferences to maximize the yield or make the quality better, or they can choose a compromise scheme.

## 4. Discussion

In order to optimize the lower irrigation limit automatic drip irrigation system in the *Lycium barbarum* field and handle the competition relationship between yield and quality, we established a simulation–optimization model for *Lycium barbarum* automatic drip lower irrigation limit, which was based on the principle of water balance, the neural network, and the NSGA-III. The validity and rationality of the model were verified by two

years of experiments. The simulation models' accuracy was verified by the Nash–Sutcliffe coefficient. The result showed that the water balance model had a high Nash–Sutcliffe coefficient, indicating this model's accuracy was good. But the ADF-50 BP-ANN model had a slightly lower Nash–Sutcliffe coefficient. According to the previous studies, the reason for this problem is the model's limited training samples [31,32]. However, it will make the expanded data tend to the mean value if the small sample expansion method is further used and make the new data lack the potential features of the original data, thus resulting in the generation of wrong samples [33]. Therefore, the accuracy of the ADF-50 BP-ANN model can only be improved by the collection of subsequent experimental data.

The model's optimization purposes were the maximum yield and the minimum ADF-50, which was an important evaluation criterion of *Lycium barbarum* quality. The non-dominated set of the optimization problem was obtained by using the multi-objective genetic algorithm NSGA-III, which included the lower limit of drip irrigation that tended to different objectives. The *Lycium barbarum* active root layer water balance model was used to calculate the irrigation quantity under different lower limits of drip irrigation. As Table 1 and Figure 9 show, irrigation times and total irrigation quantity were proportional to the irrigation lower limit, and the lower irrigation limit of scheme S2 in T1 and T3 was 10% and 5% field capacity higher than that of S3, respectively, resulted in an average increase of 13 irrigation times and 1054.19 $m^3/hm^2$ total irrigation quantity (61.5%) in the two years. Li got the same conclusion when studying the effect of irrigation limits on the water production efficiency of tomatoes [34]. And it is similar to the viewpoint that the frequency of drip irrigation is positively correlated with the lower irrigation limit, which was found by Hou when he was studying the water and heat distribution of *Lycium barbarum* orchard's soil [35]. By observing the yield and ADF-50 value of different schemes, the yield and ADF-50 increased along with the increase of total irrigation water. Similar conclusions were also obtained in other studies on *Lycium barbarum* in the same area [36]. Compared with S3, S2 increased yield by 44.2% and ADF-50 by 43.9% (The smaller ADF-50, the better quality of *Lycium barbarum*), further verifying the competitive relationship between yield and ADF-50 objectives. It can be seen that the simulation–optimization model selected the scheme which tended to yield objectives with higher irrigation lower limits so as to increase the irrigation quantity and yield; When selecting the scheme which tended to the quality of *Lycium barbarum*, the scheme with lower irrigation limit was a priority, which reduced the total irrigation quantity, yield, and value of ADF-50. When the simulation–optimization model selected a scheme that tended to one objective, it also considered this scheme's performance of the other objective. Compared with the original scheme, the scheme which tended to yield objective increased the yield by 32.6% on average during the two years of the simulation experiment but only increased the ADF-50 by 4.6%, and the quality of *Lycium barbarum* was only slightly reduced. The scheme which tended to the ADF-50 objective reduced the yield by 8.1% while reducing ADF-50 by 26.8%. The results show that NSGA-III can deal with the relationship between multiple competing objectives well. Liu and Hou also get the same conclusion when using NSGA-III to solve the multi-objective optimization problem [37,38]. S1 was a compromise scheme, which meant a compromise between the two objectives. Therefore, the lower limits of irrigation of this scheme were between S2 and S3, which were 65%, 50%, and 65% of field capacity at T1, T2, and T3. This is similar to the *Lycium barbarum's* lower irrigation limit scheme, which Xu selected by manual comparison (65%, 65%, and 55% of field capacity for the three stages) [39]. Irrigation times, the total amount of irrigation water, yield, and ADF-50 of S1 are all between S2 and S3. Compared with the original scheme, the yield of S1 increased by 10.7%, and ADF-50 decreased by 8.8%, which indicates that the simulation–optimization model can effectively improve the yield and quality of crops. Numerous scholars reached a similar conclusion when they used the simulation–optimization model to optimize the allocation of irrigation water [40–42].

In future research, we will continue to collect the irrigation quantity and ADF-50 data so as to expand the training samples of the ADF-50 BP-ANN model and improve the

model's accuracy. In addition, we will try to combine the ModelFlow model, which can simulate the groundwater flow process [43] into the simulation–optimization model so as to improve the simulation accuracy of soil water content in the *Lycium barbarum* planting area.

## 5. Conclusions

After two years of experimental simulation, the water balance model and the ADF-50 BP-ANN model can accurately simulate the change of soil water content and the mathematical relationship between ADF-50 and irrigation quantity in the *Lycium barbarum* planting area. The optimization model based on the NSGA-III algorithm can deal with the competition between the quality and yield of *Lycium barbarum* well and get the lower limit scheme of automatic drip irrigation, which tends to different objectives or compromises between two objectives. The findings of this study will effectively contribute to the planning of *Lycium barbarum's* automatic drip irrigation lower limit, and they are also very useful to other *Lycium barbarum* plant areas having similar weather conditions.

**Author Contributions:** Conceptualization, J.Z. and Y.Y.; methodology, J.Z. and Y.Y.; software, J.Z.; validation, J.L. (Jinyang Lei) and J.L. (Jun Liu); investigation, J.Z.; resources, Y.Y.; data curation, J.L. (Jinyang Lei) and J.L. (Jun Liu); writing—original draft preparation, J.Z.; writing—review and editing, J.Z. and Y.Y.; visualization, J.L. (Jinyang Lei) and J.L. (Jun Liu); supervision, Y.Y.; project administration, Y.Y.; funding acquisition, Y.Y. All authors have read and agreed to the published version of the manuscript.

**Funding:** This research was funded by [Intelligent canal and irrigation system control technology] grant number [SF-202208]. And The APC was funded by [Intelligent canal and irrigation system control technology].

**Data Availability Statement:** The data presented in this study are available on request from the corresponding author. The data are not publicly available due to it will be used for the development of subsequent models, which involves our own software patents.

**Acknowledgments:** We thank Li, Z. for experiment assistance.

**Conflicts of Interest:** The authors declare that the publication of this paper has no conflict of interest.

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
