# Peer review of "Multi-Objective Lower Irrigation Limit Simulation and Optimization Model for Lycium Barbarum Based on NSGA-III and ANN"

_water, doi:10.3390/w15040783_

Round 1

Reviewer 1 Report

The paper presents optimization process of irrigation management of  Lycium barbarum  in Ningxia Zhong-wei Jiusheng Agricultural Park  using an neural network and multi-objective genetic algorithm model NSGA-III. The work is tinteresting and has applicability in optimisation water using for irrigation. I have minor comments:
1. Please show map of experimental area
2. How did the reference crop exfoliation asses in eq 1?
3. Eq 2 - accordnig to cited reference [9], the yield was asses on rice plantation. What was reason used the same equation in this work?
4. Results: The errors between the output of water balance
model and the measured value and calculated results and the
measured values  are mean value? Why have other measures like Nash-Suctlife coefficient not used for assesment of quality models? The measure can gives criteria of quality models, like acceptable ot not acceptable.
5. Please explain T1, T2 and T3 in table 1
6. Discussion is too short. The resulta should be compare to other studies i nsimilar regian or the same plant

Author Response

We really appreciate your efforts in reviewing our manuscript. We have revised the manuscript accordingly. Our point-by-point responses are detailed below.

Q1. Please show map of experimental area.
Response:Thank you very much for your circumspection. We have added a map in line 103.

Q2.  How did the reference crop exfoliation asses in eq 1?
Response:In this study, Penman Monteith model recommended by Food and Agriculture Organization of the United Nations (FAO) was used to calculate the reference crop exfoliation, and we explained that in the line 116.

Q3. Eq 2 - accordnig to cited reference [9], the yield was asses on rice plantation. What was reason used the same equation in this work?

Response:I'm sorry for this mistake. We mislabeled the quoted article, the Eq 2 is according to cited reference [29].

Q4. Results: The errors between the output of water balance model and the measured value and calculated results and the measured values are mean value? Why have other measures like Nash-Suctlife coefficient not used for assesment of quality models? The measure can gives criteria of quality models, like acceptable or not acceptable.

Response:We are grateful for your suggestion. We used the mean value to calculate the error. However, we think it is better to use Nash efficiency formula, so we introduced Nash formula in line 126, and added the content of using Nash efficiency formula to evaluate the simulation accuracy of each model in line 277.

Q5.  Please explain T1, T2 and T3 in table 1.

Response:We agree with the comment and explained the T1, T2 and T3 in table 1. (line 310)

Q6.  Discussion is too short. The result a should be compare to other studies in similar region or the same plant.

Response:We are grateful for the suggestion. Firstly, we rewrote the discussion and described the conclusions of the paper in more detail.(line 392 to 452). To be more clear and in accordance with the reviewer concerns, we have added a brief description to compare similar study (line 419 and 438)

Thank you for your careful review. We really appreciate your efforts in reviewing our manuscript during this unprecedented and challenging time. We wish good health to you, your family, and community. Your careful review has helped to make our study clearer and more comprehensive.

Reviewer 2 Report

1-Abstract: write plant name italic and add authority with it.

2-overall revise abstract and give numeric values of results outcomes

3-Introdcution: add family and medicinal use values of crop/plant

4-correct references style/numbering mode [1][2][3], and[8][9][10].

5-Introduction lacks use of refered models in the said paper.

6-Objectives revise and make prcise

7-Results section needs formating..revise all

8-Discussion: needs revision with proper citation of the current results to the published past work on the same plant

9-Conclusion keep separate and that will give better presentation of paper

10-References cross in text and relevant part/section.

Author Response

We really appreciate your efforts in reviewing our manuscript. We have revised the manuscript accordingly. Our point-by-point responses are detailed below.

Q1. Abstract: write plant name italic and add authority with it.

Response:Thank you for reminding me. We have rewritten the plant name in italic type and added authority with it.(Line 10).

Q2. overall revise abstract and give numeric values of results outcomes.

Response:We have overall revised abstract and given numeric values of results outcomes.(Line 20-29)

Q3. Introdcution: add family and medicinal use values of crop/plant.

Response:We added the this content in Line 38.

Q4. correct references style/numbering mode [1][2][3], and[8][9][10].

Response:We have corrected reference style.

Q5. Introduction lacks use of referred models in the said paper.

Response:Thank you for your suggestion. We have added an introduction to the application of these models in the previous studies. (Line 60-98)

Q6. Objectives revise and make precise.

Response:We have rewritten the objectives. It is worth mentioning that ANN can not give an obvious functional relationship, but can only give a potential calculation function, so fANN(x) is used to represent the calculation formula. (Line 241)

Q7. Results section needs formating..revise all.

Response:We have rewritten results according to the reviewer’s suggestion for clearer expression. (Line 308-369 )

Q8. Discussion: needs revision with proper citation of the current results to the published past work on the same plant.

Response:We are grateful for the suggestion. Firstly, we rewrote the discussion and described the conclusions of the paper in more detail.(Line 392 to 452). To be more clear and in accordance with the reviewer concerns, we have added a brief description to compare similar study (Line 419 and 438)

Q9. Conclusion keep separate and that will give better presentation of paper.

Response:Thank you for your suggestion. We have separated the conclusion and rewritten this part to present our conclusions more clearly.

Q10. References cross in text and relevant part/section.

Response: There is indeed a cross-reference problem in one of the articles, and we have corrected it. If the problem still exists, there may be a problem generating the PDF file during the upload process. I will consult the editor about this problem.

Thank you for your careful review. We really appreciate your efforts in reviewing our manuscript during this unprecedented and challenging time. We wish good health to you, your family, and community. Your careful review has helped to make our study clearer and more comprehensive.

Reviewer 3 Report

Dear Authors,

I appreciate your article entitled “Multi-objective Lower Irrigation Limit Simulation and Optimization Model for Lycium Barbarum Based on NSGA-â…¢ and ANN”, however I ask a little more effort to make it more clear and robust. I tried to provide some general and specific suggestions that I hope will help you.

Introduction: please try to expand the references and highlight better the novelty of the work, now is not so clear for me.

Materials and methods: In the presentation of the study area maybe it could help to add a map. Please add some more theoretical references, some definitions, etc. would not hurt and, above all, would make the article easily understandable even to a non-professional (not everyone knows these techniques).

Discussion and conclusions: The part of discussion should be expanded, even by adding more bibliographic references. Moreover, the limits of the model that you mention in the conclusions, in my opinion, should be better described in the methodological part, it is an approximation of water that should be clearly highlighted.

Please correct the format of the bibliography numbering, it is: 1. not [1]. In general, please also check the editing of the references.

The part relating to author contributions is missing.

Please find below some specific suggestions:

·         Please define the acronyms, for example at line 17 NSGA-III, or at line 135: What is BP?

·         Please check upper and lower cases

·         Please adjust the bibliographic references in the text, they are not quotes and then you have to do [1-3] no [1][2][3]

·         Please use separator for thousands when indicate numbers larger than 3 digits

·         Line 110: h should be in italics.

·         Line 110: Pi should be Pi, the same for line 111.

·         Please check for spaces among words and among lines of text and figures or tables.

·         Lines 200, 207, 217: Are they headings in subparagraphs? It is not understandable.

·         Fig. should be Figures and Tab should be Table

·         Line 304-307: it is hard to read the results, please find another way to “describe” figure 8

Author Response

We really appreciate your efforts in reviewing our manuscript. We have revised the manuscript accordingly. Our point-by-point responses are detailed below.

Q1. Introduction: please try to expand the references and highlight better the novelty of the work, now is not so clear for me.

Response:Thank you for your suggestions. We have expand the references in introduction. And we rewrote this section to emphasize the innovative content of this study (line 60-98).

Q2. Materials and methods: In the presentation of the study area maybe it could help to add a map. Please add some more theoretical references, some definitions, etc. would not hurt and, above all, would make the article easily understandable even to a non-professional (not everyone knows these techniques).

Response:Thanks to Reviewer for reminder, we added a map in line130 and some theoretical references (line 175,176) for making the article easily understandable.

Q3. Discussion and conclusions: The part of discussion should be expanded, even by adding more bibliographic references. Moreover, the limits of the model that you mention in the conclusions, in my opinion, should be better described in the methodological part, it is an approximation of water that should be clearly highlighted.

Response:Thank you for the suggestion. We have expanded the discussion and added more references. In addition, we have moved the content of  models’ limits to the methodological part (Line 133 and 183).

Q4. Please correct the format of the bibliography numbering, it is: 1. not [1]. In general, please also check the editing of the references.

Response:Thank you for your circumspection. We have corrected the numbering and check all references.

Q5. The part relating to author contributions is missing.

Response:According to the reviewer’s comment, we have added author contributions according to the reviewer’s comment.(Line 464)

Q6. Please find below some specific suggestions:

  • Please define the acronyms, for example at line 17 NSGA-III, or at line 135: What is BP?
  • Please check upper and lower cases.
  • Please adjust the bibliographic references in the text, they are not quotesand then you have to do [1-3] no [1][2][3].
  • Please use separator for thousandswhen indicate numbers larger than 3 digits
  • Line 110: h should be in italics.
  • Line 110: Pi should be Pi, the same for line 111.
  • Please check for spaces among words and among lines of text and figures or tables.
  • Lines 200, 207, 217: Are they headings in subparagraphs? It is not understandable.
  • Fig. should be Figures and Tab should be Table.
  • Line 304-307: it is hard to read the results, please find another way to “describe”figure 8.

Response:Thank you for your valuable comment, we have defined all the acronyms in line 19 and 175. We corrected upper and lower cases and the reference in line 203,205. We also used separator for thousands and checked all spaces among lines of text and figures or tables. Then we rewrote the parameter and sub paragraphs’ heading in right way (Line143,243,250,259). According the reviewer’s comment,we Finally we changed the Fig. and Tab to Figure and Table and rewrote the description of figure 8 (line 353-359).

Thank you for your careful review. We really appreciate your efforts in reviewing our manuscript during this unprecedented and challenging time. We wish good health to you, your family, and community. Your careful review has helped to make our study clearer and more comprehensive.
